# Molecular Characterization of *Staphylococcus aureus* Complex Isolated from Free-Ranging Long-Tailed Macaques at Kosumpee Forest Park, Maha Sarakham, Thailand

**DOI:** 10.3390/tropicalmed8070374

**Published:** 2023-07-20

**Authors:** Natapol Pumipuntu, Tawatchai Tanee, Penkhae Thamsenanupap, Pensri Kyes, Apichat Karaket, Randall C. Kyes

**Affiliations:** 1One Health Research Unit, Mahasarakham University, Maha Sarakham 44000, Thailand; 2Veterinary Infectious Disease Research Unit, Mahasarakham University, Maha Sarakham 44000, Thailand; 3Faculty of Veterinary Sciences, Mahasarakham University, Maha Sarakham 44000, Thailand; 4Faculty of Environment and Resource Studies, Mahasarakham University, Maha Sarakham 44150, Thailand; 5Department of Psychology, Center for Global Field Study and Washington National Primate Research Center, University of Washington, Seattle, WA 98195, USA; 6Department of National Parks, Wildlife and Plant Conservation, Bangkok 10900, Thailand; 7Departments of Psychology, Global Health, Anthropology and Center for Global Field Study, Washington National Primate Research Center, University of Washington, Seattle, WA 98195, USA

**Keywords:** *Staphylococcus aureus*, MRSA, MSSA, *Staphylococcus argenteus*, long-tailed macaques, molecular characteristic, virulence genes

## Abstract

The *Staphylococcus* (*S*.) *aureus* complex, including methicillin-resistant *S. aureus* (MRSA) and methicillin-susceptible *S. aureus* (MSSA), and *S. argenteus* are bacterial pathogens that are responsible for both human and animal infection. However, insights into the molecular characteristics of MRSA, MSSA, and *S. argenteus* carriages in wildlife, especially in long-tailed macaques, rarely have been reported in Thailand. The objective of this study was to assess molecular characterization of MRSA, MSSA, and *S. argenteus* strains isolated from free-ranging long-tailed macaques (*Macaca fascicularis*) at Kosumpee Forest Park, Maha Sarakham, Thailand. A total of 21 secondary bacterial isolates (including 14 MRSA, 5 MSSA, and 2 *S. argenteus*) obtained from the buccal mucosa of 17 macaques were analysed by a Polymerase chain reaction (PCR) to identify several virulence genes, including *pvl*, *tst*, *hla*, *hlb clfA*, *spa* (x-region), *spa* (IgG biding region), and *coa*. The most prevalent virulence genes were *clfA*, *coa*, and the *spa* IgG biding region which presented in all isolates. These data indicated that MRSA, MSSA, and *S. argenteus* isolates from the wild macaques at Kosumpee Forest Park possess a unique molecular profile, harbouring high numbers of virulence genes. These findings suggest that wild macaques may potentially serve as carriers for distribution of virulent staphylococcal bacteria in the study area.

## 1. Introduction

The *Staphylococcus aureus* complex is an opportunistic bacterial pathogen capable of colonizing a diverse range of animal species, including humans, causing severe infections [1]. It poses a zoonotic risk, as it can spread among environments, humans, and various animals, such as pets, livestock, and wildlife [2]. Methicillin-resistant *Staphylococcus aureus* (MRSA) is a drug-resistant strain of *S. aureus*, exhibiting resistance to methicillin and other antibiotics through the expression of a penicillin-binding protein that confers resistance to β-lactam antibiotics [3]. MRSA has emerged as a dangerous antimicrobial-resistant pathogen capable of colonizing humans, domestic animals, livestock, and wildlife, causing acute and chronic infections in both animal and human hosts [4]. It differs from methicillin-susceptible Staphylococcus aureus (MSSA) due to its resistance to β-lactam antibiotics, which is mediated by the *mecA* gene carried on the Staphylococcal Cassette Chromosome mec (SCCmec). This genetic element encodes an alternative penicillin-binding protein, namely, PBP-2a, which alters the target site for β-lactam antibiotics. The majority of clinical MRSA isolates (90%) carry the *mecA* gene on their SCCmec [5,6]. The main mechanism of drug resistance involves a significantly decreased affinity of β-lactam antibiotics for this altered substrate [7]. MRSA is classified into three groups: hospital/healthcare-associated MRSA (HA-MRSA), community-associated MRSA (CA-MRSA), and livestock-associated MRSA (LA-MRSA). HA-MRSA is associated with nosocomial and surgical wound infections, leading to treatment failure and increased morbidity and mortality. CA-MRSA typically causes various infections in humans, including sepsis, pneumonia, skin infections, and soft tissue infections. LA-MRSA is specifically associated with animal infections. MRSA has been shown to spread epidemiologically in both human communities and animal populations, including companion animals, livestock, and wildlife [8].

A number of studies have reported that *S. aureus* isolated from non-human primates had close similarity with human isolates, leading to the concern of amphixenoses or zoonotic diseases that can be transmitted from humans to other species (e.g., non-human primates) and vice versa. [2,9,10]. Among non-human primates, an increasing number of studies have reported the presence of MRSA, including in captive gorillas in the USA [11], chimpanzees in Africa [10,11], wild rhesus macaques in Nepal [12], and, most recently, in wild long-tailed macaques in Thailand [2]. Current research indicates that MRSA can be transmitted from animals to humans and vice versa [10,13].

The *S. aureus* complex possesses numerous virulence genes and produces a diverse range of virulence factors that contribute to its pathogenicity [14,15]. Key virulence factors include coagulase, protein A, hemolysin, clumping factors, toxic shock syndrome toxin-1, and Panton–Valentine leucocidin [14,15,16]. Coagulase promotes clotting by interacting with soluble fibrinogen, leading to abscess formation and bacterial persistence in host tissues [17]. Protein A and the IgG-binding protein facilitate immunoglobulin precipitation by binding the Fc region, masking surface antigens that inhibit opsonization and phagocytic killing by immune cells [18,19]. Hemolysins lyse red blood cells and host cells to acquire nutrients [20]. Clumping factors enable attachments to host epithelial cells [21]. Panton–Valentine leukocidin selectively kills phagocytic cells, resulting in tissue necrosis [22].

At present, there are very few studies examining the molecular characteristics of the *S. aureus* complex (both MRSA and MSSA) and *S. argenteus* in free-ranging wildlife (including macaques), especially in Thailand. Understanding of the virulence genes’ carriage is essential for accessing the severity of the pathogens in wildlife and developing appropriate strategies to mitigate and prevent this infectious disease transmission. The purpose of the present study was to identify the molecular characteristics of MRSA, MSSA, and *S. argenteus* by identifying and investigating the prevalence of virulence genes, including *pvl* (Panton–Valentin leukocydin), *tst* (toxic shock syndrome toxin-1), *hla* (alpha hemolysin), *hlb* (beta hemolysin), *clfA* (clumping factor A), *spa* (x-region of staphylococcal protein A), *spa* (IgG biding region of staphylococcal protein A), and *coa* (coagulase) in MRSA, MSSA, and *S. argenteus* isolated from buccal swab of free-ranging long-tailed macaques at Kosumpee Forest Park, Maha Sarakham, Thailand.

## 2. Materials and Methods

### 2.1. Sample Collection and Preparation

Bacterial isolates from the buccal cavities of wild long-tailed macaques were collected as part of our previous work focusing on the human–primate conflict and coexistence at Kosumpee Forest Park (KFP) [2,23]. Sampling procedures, the study site, and the KFP macaque population have been detailed in our previous studies [2,23,24,25]. Buccal swab samples were collected between 10–11 November 2018 from 30 randomly selected wild long-tailed macaques living around KFP (16°15′12.6″ N 103°04′02.0″ E) in Maha Sarakham province, Northeastern Thailand (Figure 1). The macaques were captured using a nylon mesh trapping cage, measuring 1.50 × 0.40 × 0.40 m. Tiletamine–zolazepam (Zoletil^®^ 100 mg/mL, Virbac, Carros, France) was delivered intramuscularly (via blowpipe) to anesthetize the macaques in the cage. Once sedated, the macaques were retrieved from trapping cage for sampling. Their vital signs were monitored through the sampling protocol until recovery and subsequent release back to their groups. Buccal samples were collected following aseptic techniques involving swabbing of the macaques’ buccal cavity using sterile collection swabs composed with transport media (Deltalab, Bacelona, Spain). All trapping and sampling procedures were supervised and conducted by wildlife veterinary specialists from Mahasarakham University (MSU) according to the protocol and approval by the Institutional Animal Care and Use Committee at Mahasarakham University and the Thai Department of National Parks (see ethics statement at Institutional Review Board Statement part). Upon collection, all buccal swab samples were kept at 4 °C and transferred to the One Health Research Unit (OHRU), Veterinary Public Health laboratory of the Faculty of Veterinary Sciences at Mahasarakham University, for bacteriological analysis.

Thirty buccal swab samples were prepared for bacterial culture using Baird–Parker Agar supplemented with egg yolk and potassium tellurite (Oxoid, Hampshire, UK), a selective media for analysing *S. aureus*. Cultures were incubated at 37 °C for 24 h. Suspected bacterial colonies were identified for *S. aureus* using conventional methods such as Gram staining, the catalase enzyme test, mannitol salt agar selectivity (MSA), deoxyribonuclease (DNase) test, tube coagulase test (TCT), and confirmed by a rapid agglutination test, which was accomplished using the Staphaurex™ Plus latex agglutination test (Thermo Fisher Scientific, Waltham, MA, USA). *S. aureus* ATTC 25923 was used as a positive control strain for conventional methods of bacterial identification.

Genomic DNA of all bacterial isolates was extracted using a DNA extraction kit (Geneaid, Taiwan) following the protocol for Gram-positive bacteria. As described in our previous work [2], methicillin-resistant *S. aureus* (MRSA) and *S. argenteus* isolates were determined by PCR amplification method for the presence of the *mecA* gene to confirm MRSA and the *NRPS* gene to differentiate *S. argenteus*.

### 2.2. Detection of Staphylococcal Virulence Genes

Genomic DNA of all bacterial isolates was quantified and measured via the OD 260/280 nm ratio with the NanoDrop 1000 Spectrophotometer (Thermo Scientific, Branchburg, NJ, USA) and stored in microtubes at −20 °C until molecular identification. All *Staphylococcus* spp. bacterial DNA were amplified for eight virulence genes: *tst*, *coa*, *spa* (x-region), *spa* (IgG-binding region), *hla, hlb, clfA*, and *pvl*. Amplification was achieved with specific oligonucleotide primers, as shown in Table 1.

All PCR reaction mixtures contained a total volume of 25 μL and comprised 10 mM of each forward and reverse primer, 0.2 mM dNTPs, 2 mM MgCl_2_, 1 U of Taq DNA polymerase, 1× Taq reaction buffer, and 100 ng DNA template. PCR was performed using a thermocycler (Biometra GmbH, Jena, Germany), and the program began with an initial denaturation at 95 °C for 10 min. This was followed by 35 cycles of 95 °C for 30 s, 55 °C for 30 s and 72 °C for 30 s and a final extension at 72 °C for 10 min. PCR amplicons were subjected to 1.5% agarose gel electrophoresis and staining with ViSafe Red Gel Stain (Vivantis, Malaysia). The targeted PCR products were observed and visualized for their DNA bands under ultraviolet light by a Gel Doc XR+ Documentation System (Bio-Rad, Hercules, CA, USA). The control bacterial template for the PCR included the strains ATCC 19095 (*tst*) and ATCC13565 (*coa*, *clfA*, *hla*, *hlb*, *spa* x-region, IgG-binding region). For *pvl*, the PCR product was selected for nucleotide sequencing, and we analysed their sequence bases to confirm (Accession number: AB084255.1).

### 2.3. Statistical Analysis

Descriptive statistics were used to define the prevalence of virulence genes of MRSA, MSSA, and *S. argenteus*. The Chi-square test and Fisher’s exact test of independence were used to analyse the difference between virulence genes detected for each bacterial group in the SPSS statistics program (version 22). Values were considered statistically significance with a probability value (*p*-value) ≤ 0.05.

## 3. Results

A total of 21 bacterial isolates (14 MRSA, 5 MSSA, and 2 *S. argenteus*) from 17 macaques were analysed by a Polymerase chain reaction (PCR) to test for the presence of virulence genes, including *pvl, tst, hla, hlb clfA, spa* (x-region), *spa* (IgG biding region), and *coa*, as shown in Figure 2. The most prevalent staphylococcal virulence genes for all bacterial groups were *clfA*, *coa*, and *spa* (IgG biding region), which were present in all bacterial isolates (100%; 21/21), followed by *spa* (x-region) (95.24%; 20/21).

Staphylococcal virulence genes such as *tst* and *pvl* were highly prevalent at 57.14% (12/21) and 61.9% (13/21), respectively. For the hemolysin gene, 15 (85.71%) of the isolates carried *hla*, which was the most prevalent hemolysin gene in this study. Additionally, 12 (71.43%) of the isolates carried *hlb*. Interestingly, all bacterial isolates carried at least one type of haemolysis gene, and 11 isolates (52.38%) carried both *hla* and *hlb* genes, as shown in Table 2. In addition, a comparative analysis of the prevalence of virulence genes among MRSA, MSSA, and *S. argenteus* using a Chi-square test and Fisher’s exact test revealed no significant differences in the likelihoods of those bacterial pathogens harbouring each virulence gene (*p*-value > 0.05).

## 4. Discussion

The prevalence of the major virulence genes of MRSA, MSSA, and *S. argenteus* in buccal swabs of long-tailed macaques was examined in this study. The results indicated the presence of eight virulence genes that were major virulence indicators linked to the staphylococcal pathogenicity of the bacterial samples obtained from the long-tailed macaques. These findings also were similar to previous studies that isolated bacteria from animal origins [15,32]. Several genes in this study (*hla*, *hlb, tst*, and *pvl*) also have been detected in aggressive isolates of *S. aureus* [33,34]. These virulence genes spread through the isolates at a high frequency (57.14–100%) after amplification by PCR. As such, it can be inferred that the MRSA, MSSA, and *S. argenteus* isolated from the long-tailed macaques in this study may have had high pathogenicity. Furthermore, the majority of the isolates in our study demonstrated a variety of virulence gene combinations, demonstrating that these bacterial samples had considerable genetic diversity.

It is well known that *S. aureus* produces TSST-1 that has been reported to be associated with toxic shock syndrome [35] and produces systemic infections [36]. Surveillance for the prevalence of *tst*, which encodes the pyrogenic toxin superantigen TSST-1 in *S. aureus* isolates from the oral carriage of the wild macaques, is very important from a One Health standpoint since it could provide information regarding the status of staphylococcal virulence species circulating in potential wildlife reservoirs in the study area. The *tst* gene is located on the pathogenicity islands (PAIs) of staphylococcal bacteria, which assists the immunopathogenesis of *S. aureus*/*S. argenteus* via the anti-inflammatory chemokines of the host and also facilitates the induction of immunosuppression [37]. Consistent with the results of other studies, this gene has been reported to be detected in *S. aureus* (both MRSA and MSSA) isolates from wildlife reservoirs, including wild avians, red deer, wild rodents, boars, and also non-human primates [36]. Our results indicated that *S. aureus* carrying the toxic shock syndrome toxin gene is widely distributed throughout the macaque population at the forest park. These bacteria possess the ability to produce toxic proteins that may be significant in the pathogenicity of infectious disease for animals and can be harmful to humans who become infected from the animal reservoirs.

In this study, we found a moderate to high prevalence of the *pvl* gene, which encodes for Panton–Valentine leucocidin in MRSA, MSSA, and *S. argenteus*. It has not been reported in wildlife reservoirs in Thailand to date. Nonetheless, MRSA and MSSA isolates carrying the *pvl* gene have been reported in wild rodents and non-human primates from Germany, Portugal, Czech, and France [36]. Furthermore, some studies have reported that most of the *S. aureus* isolates carrying the *pvl* gene derived from wildlife carriages were frequently detected from non-human primates [36,38]. These reports indicate that *S. aureus* harbouring the *pvl* gene derived from wild monkeys may play a role as natural maintenance hosts for pathogenic *S. aureus* (both MRSA and MSSA), which can affect human health. Panton–Valentine leucocidin is a significant pore-forming cytotoxin that is regularly associated with abscesses in hosts and plays a key role in the induction of leukocyte destruction and severe necrotic lesions of soft tissues and skin [39]. This cytotoxin has a high potential virulence and could be responsible for clinical manifestations, as reported in a previous study [40].

Several addition genes encoding general virulence factors of *S. aureus* (MRSA and MSSA) and *S. argenteus* were identified, including *coa*, *hla*, *hlb*, *clfA*, *spa* (x region), and *spa* (IgG-binding region). All bacterial isolates in this study harboured *coa*, *clfA*, and *spa* (IgG-binding region). The *spa* is known to be the fundamental virulence gene for *S. aureus*, regarding disease development and severity by encoding staphylococcal protein A [15]. Protein A has an ability as a bacterial trait to precipitate immunoglobulins, mask underlying surface antigens, and inhibit opsonization and phagocytic killing of staphylococci by PMN cells [41]. For the Coagulase protein encoded from *coa* genes, it is secreted from *S. aureus* to promote coagulation, which leads to the formation of abscesses and bacterial persistence in host tissues [17]. Expression of the *clfA* gene that encodes clumping factors A of *S. aureus* is thought to enhance bacterial growth and promote infection in the face of host defence mechanisms, such as phagocytosis [42]. Furthermore, both alpha and beta hemolysins encoded from *hla* and *hlb* genes, respectively, can damage platelets and destroy lysosomes, in turn, causing necrosis and ischemia in the host [20].

Our study of the molecular characterization of *S. aureus* complex (both MRSA and MSSA) and *S. argenteus* indicate that these bacteria have the potential to be aggressive pathogens among wild long-tailed macaques in Kosumpee Forest Park, Thailand. This indicates a need for vigilance regarding wildlife reservoirs of these bacterial zoonoses. Given their pathogenic potential to cause severe disease in both animals and humans, a proactive approach to veterinary public health risk management is essential to protect the people and animals who may come into contact with the infected animals in this study area.

## 5. Conclusions

The wild long-tailed macaques at Kosumpee Forest Park may serve as a potential reservoir of *S. aureus* (MRSA and MSSA) and *S. argenteus.* Based on the molecular characterization of these Staphylococci bacteria in this study, the macaques were found to carry antimicrobial resistance pathogens and virulence strains that could potentially affect other animals and humans. Given the genetic diversity of MRSA, MSSA, and *S. argenteus* in wild monkey reservoirs and the close interactions between the people and macaques at Kosumpee Forest Park, regular monitoring of those pathogens is needed for effective risk prevention and intervention.

## Figures and Tables

**Figure 1 tropicalmed-08-00374-f001:**
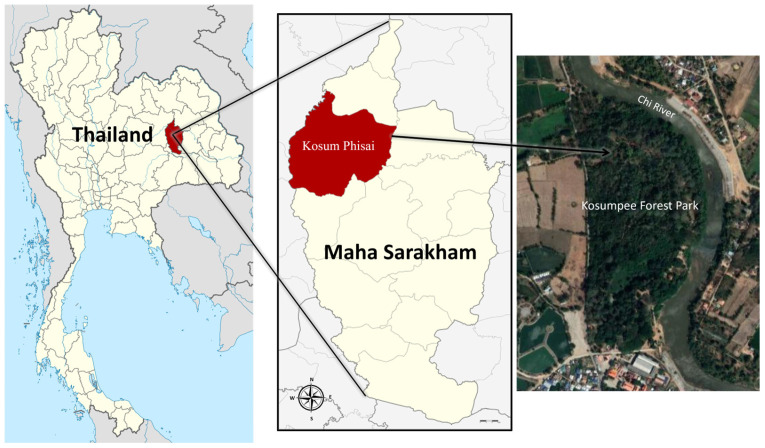
Location of the study area at Kosumpee Forest Park, located in Kosum Phisai district, Maha Sarakham province, in Northeast Thailand.

**Figure 2 tropicalmed-08-00374-f002:**
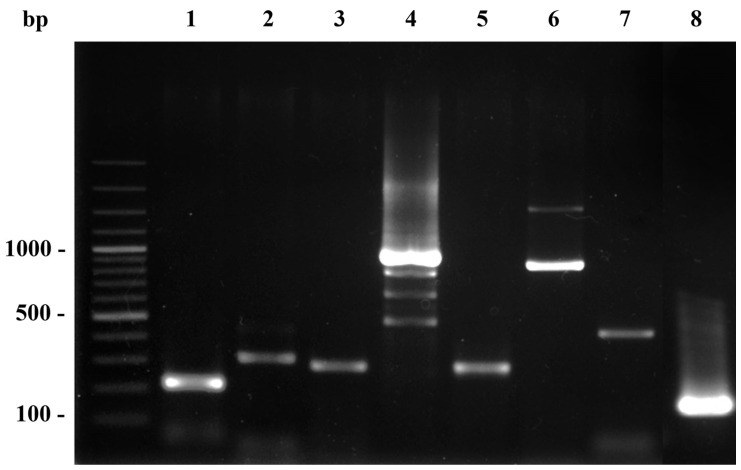
PCR amplicons of virulence genes of staphylococcal bacteria. Lane M; DNA molecular size ladder; lane 1–8, PCR amplicons of *hla* (209 bp), *hlb* (309 bp), *clfA* (229 bp), *coa* (pleomorphism 410, 740, 910, 970 bp), *spa* x-region (320 bp), *spa* IgG-biding region (920 bp), *pvl* (433 bp), and *tst* gene (180 bp) respectively.

**Table 1 tropicalmed-08-00374-t001:** Specific oligonucleotide primers for amplification of *Staphylococcus* spp. virulence genes.

Target Gene	Sequence (5′–3′)	Amplicon Size (bp)	Ref.
*tst*	F: TTCACTATTTGTAAAAGTGTCAGACCCACT R: TACTAATGAATTTTTTTATCGTAAGCCCTT	180	[26]
*coa*	F: CGAGACCAAGATTCAACAAG R: AAAGAAAACCACTCACATCA	410, 740, 910, 970	[27]
*clfA*	F: ATTGGCGTGGCTTCAGTGCT R: CGTTTCTTCCGTAGTTGCATTTG	292	[28]
*hla*	F: CTGATTACTATCCAAGAAATTCGATTG R: CTTTCCAGCCTACTTTTTTATCAGT	209	[29]
*hlb*	F: GTGCACTTACTGACAATAGTGC R: GTTGATGAGTAGCTACCTTCAGT	309	[29]
*spa* (x-region)	F: CAAGCACCA AAAGAGGAA R: CACCAGGTTTAACGACAT	320	[30]
*spa* (IgG-biding region)	F: CACCTGCTGCAAATGCTGCG R: GGCTTGTTGTTGTCTTCCTC	920	[31]
*pvl*	F: ATCATTAGGTAAAATGTCTGGACATGATCCA R: GCATCAASTGTATTGGATAGCAAAAGC	433	[32]

**Table 2 tropicalmed-08-00374-t002:** Number of virulence genes detected in MRSA, MSSA, and *S. argenteus* isolated from macaque buccal swabs.

Macaque ID	Isolate No.	Bacterial Identification	Virulence Genes Detection
M 1	1.1	MRSA	*clfA, coa, spa (Ig)*, *spa(X), hlb*, *tst*, *pvl*
	1.2	MSSA	*clfA, coa, spa (Ig)*, *spa(X), hla, hlb*, *pvl*
M 2	2.2	MRSA	*clfA, coa, spa (Ig)*, *spa(X), hla*, *tst*, *pvl*
M 4	4.1	MRSA	*clfA, coa, spa (Ig)*, *spa(X), hla*, *tst*
M 7	7.3	MRSA	*clfA, coa, spa (Ig)*, *spa(X), hla*, *hlb*, *tst*
M 9	9.1	MRSA	*clfA, coa, spa (Ig)*, *spa(X), hla*, *hlb*, *tst*
M 15	15.1	MRSA	*clfA, coa, spa (Ig)*, *spa(X), hla*, *hlb*, *tst*
M 16	16.1	MRSA	*clfA, coa, spa (Ig)*, *spa(X), hla*, *hlb*
M 17	17.1	*S. argenteus*	*clfA, coa, spa (Ig)*, *spa(X), hla, pvl*
M 19	19.2	MSSA	*clfA, coa, spa (Ig)*, *spa(X), hla*, *pvl*
M 21	21.3	MSSA	*clfA, coa, spa (Ig)*, *spa(X)*,
	21.1	MRSA	*clfA, coa, spa (Ig)*, *spa(X), hla, hlb*, *tst*, *pvl*
M 22	22.1	MSSA	*clfA, coa, spa (Ig)*, *spa(X), hla*, *tst*
	22.4	MRSA	*clfA, coa, spa (Ig)*, *hla*
M 24	24.1	MRSA	*clfA, coa, spa (Ig)*, *spa(X), hla, hlb*, *pvl*
M 25	25.1	MRSA	*clfA, coa, spa (Ig)*, *spa(X), hla*, *hlb*, *tst*, *pvl*
M 26	26.5	MSSA	*clfA, coa, spa (Ig)*, *spa(X), hla*, *hlb*, *pvl*
M 27	27.3	MRSA	*clfA, coa, spa (Ig)*, *spa(X), hla*, *hlb*, *tst*, *pvl*
M 28	28.1	MRSA	*clfA, coa, spa (Ig)*, *spa(X), hla*, *hlb*, *tst*, *pvl*
	28.4	*S. argenteus*	*clfA, coa, spa (Ig)*, *spa(X), hlb, pvl*
M 29	29.5	MRSA	*clfA, coa, spa (Ig)*, *spa(X), hla*, *hlb*, *pvl*

spa (X) = spa x-region; spa (Ig) = spa IgG biding region.

## Data Availability

Not applicable.

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
