# Peer review of "Molecular Characterization of Staphylococcus aureus Complex Isolated from Free-Ranging Long-Tailed Macaques at Kosumpee Forest Park, Maha Sarakham, Thailand"

_tropicalmed, 2023, doi:10.3390/tropicalmed8070374_

Round 1

Reviewer 1 Report

Dear Authors

I happily endorsed the publication of this manuscript

Author Response

Tropical Medicine and Infectious Disease

-------------------------------------------------------------------------------------------------------------------------

Title: Molecular characterization of methicillin-resistant Staphylococcus aureus, methicillin-susceptible Staphylococcus aureus, and Staphylococcus argenteus isolated from free-ranging long-tailed macaques at Kosumpee Forest Park, Maha Sarakham, Thailand

Manuscript Number: tropicalmed-2490476

Author: Natapol Pumipuntu, Tawatchai Tanee, Pensri Kyes, Penkhae Thamsenanupap, Apichat Karaket, Randall C. Kyes

-------------------------------------------------------------------------------------------------------------------------

Response to reviewers

Reviewer # 1 comments:

Comment 1: I happily endorsed the publication of this manuscript.

Authors respond: Thank you, we appreciate your kind review and comment.

Reviewer 2 Report

1-    How the studies claim their novelty is that there are many drug-resistant and susceptible bacteria that were isolated from cattle and livestock. as the manuscript itself describes them—as Staphylococcus aureus long-tailed macaques in Thailand.

2-    Are the used livestock that are associated with these isolates facing any clinical symptoms or adverse effects like morbidity, etc.? As the author claims in line 209, Our results indicated that S. aureus carrying the toxic shock syndrome toxin gene is widely distributed throughout the Ma-210 caque population at the forest park.

3-    What are the virulence factors associated with these isolates?

4-    How these bacteria differ from the previously known and described drug-resistant islates

5-    Elaborated amphixenosis transmission.

6-    The abbreviated form of virulence genes like tst, pvl, etc. needs to be elaborated.

7-    manuscript lacking the description of the drug-resistant profile of the isolate as mentioned in the title: methicillin-resistant Staphylococcus aureus, methicillin-susceptible.

8-    What is the prevalence of the hla, hlb, tst, and pvl genes in methicillin-resistant Staphylococcus aureus and methicillin-susceptible ----- In line 191, Several genes in this study (hla, hlb, tst, and pvl) have also been detected in aggressive isolates of S. aureus.

9-     In many places, the bacterial name is not italicized, and it needs to be checked and corrected in the whole manuscript.

Minor editing of English language required

Author Response

Tropical Medicine and Infectious Disease

-------------------------------------------------------------------------------------------------------------------------

Title: Molecular characterization of methicillin-resistant Staphylococcus aureus, methicillin-susceptible Staphylococcus aureus, and Staphylococcus argenteus isolated from free-ranging long-tailed macaques at Kosumpee Forest Park, Maha Sarakham, Thailand

Manuscript Number: tropicalmed-2490476

Author: Natapol Pumipuntu, Tawatchai Tanee, Pensri Kyes, Penkhae Thamsenanupap, Apichat Karaket, Randall C. Kyes

-------------------------------------------------------------------------------------------------------------------------

Response to reviewers

Reviewer # 2 comments:

Comment 1: How the studies claim their novelty is that there are many drug-resistant and susceptible bacteria that were isolated from cattle and livestock. as the manuscript itself describes them—as Staphylococcus aureus long-tailed macaques in Thailand.

Authors respond: Thank you for your comment. At present, there are many reports of MRSA isolated from cattle and livestock, especially in Thailand. However, it is worth noting that studies of MRSA in long-tailed macaques are quite limited. Moreover, assessment of the Molecular characterization of MRSA is rare, especially in Thailand as we noted in the ms.

Comment 2: Are the used livestock that are associated with these isolates facing any clinical symptoms or adverse effects like morbidity, etc.? As the author claims in line 209, Our results indicated that S. aureus carrying the toxic shock syndrome toxin gene is widely distributed throughout the Macaque population at the forest park

Authors respond: To our knowledge and from our previous studies, MRSA/MSSA and S.argenteus carrying those virulence genes can affect livestock, including dairy cows. When dairy cows are infected with these bacteria, they may exhibit a range of clinical manifestations including mastitis which is an inflammation of the udder, swollen and painful udders, changes in milk consistency, decreased milk production, and also cause skin infections in dairy cows, resulting in lesions, abscesses, or ulcers on the skin. These infections can be localized or spread over a larger area. Affected cows may exhibit signs of discomfort, itching, or pain. However, we did not detect the clinical manifestation in the macaques in this study, but, those bacteria still have a potential virulence as a result of virulence genes detection.

Comment 3: What are the virulence factors associated with these isolates?

Authors respond: From the detection of virulence genes, those bacteria can produce a range of virulence factors that can damage the host cell including coagulase enzyme, protein A (x-region and IgG-binding region), hemolysin, clumping factors, toxic shock syndrome toxin-1, and Panton-Valentin Leukocydin.

Comment 4: How these bacteria differ from the previously known and described drug-resistant isolates

Authors respond: In this study, we found a moderate to high prevalence of the pvl gene which encodes for Panton–Valentine leucocidin in MRSA, MSSA and S. argenteus which were predominantly higher than the bacterial isolated from bovine mastitis in many previous studies in Thailand. In addition, it has not been reported in wildlife reservoirs in Thailand to date. For the drug-resistant isolates description, we only detected the mecA gene that refers to methicillin resistant isolates, but we did not analyze their antimicrobial susceptibility  in our research. 

 Comment 5: Elaborated amphixenosis transmission.

Authors respond: We sincerely appreciate this comment. Amphixensis is a zoonosis disease that can be passed from humans to other species and vice versa. As is well known, MRSA is a type of antibiotic-resistant bacteria commonly found in human healthcare settings. While it primarily affects humans, it can also be transmitted to animals, particularly pets, livestock and wildlife such as long-tailed macaques, through close contact. As such, the staphylococcal infection in this study can be referred to as amphixenosis.

We also added the sentence “…or zoonotic diseases that can be transmit from humans to other species (e.g., non-human primates) and vice versa.” in line 60-62 of the revised manuscript.

Comment 6: The abbreviated form of virulence genes like tst, pvl, etc. needs to be elaborated.

Authors respond: Thank you for your advice. We accept your comment. We added the sentence “… pvl (Panton-Valentin leukocydin), tst (toxic shock syndrome toxin-1), hla (alpha hemolysin), hlb (beta hemolysin), clfA (clumping factor A), spa (x-region of staphylococcal protein A), spa (IgG biding region of staphylococcal protein A), and coa (coagulase)…” in line 83-86 of the revised manuscript according to your comment.

Comment 7: manuscript lacking the description of the drug-resistant profile of the isolate as mentioned in the title: methicillin-resistant Staphylococcus aureus, methicillin-susceptible.

Authors respond: We appreciate your valuable feedback and suggestions. According to the manuscript, we detected only mecA gene that refers to methicillin resistant isolates, but we did not analyze their antimicrobial susceptibility in our research because of the main aspects of their virulence genes characteristics. However, we apologize for any confusion caused by the title of our manuscript. We acknowledge that the title suggests the inclusion of both methicillin-resistant Staphylococcus aureus (MRSA) and methicillin-susceptible Staphylococcus aureus (MSSA) strains, and the lack of detailed description regarding the drug-resistant profile may seem inconsistent.

Comment 8: What is the prevalence of the hla, hlb, tst, and pvl genes in methicillin-resistant Staphylococcus aureus and methicillin-susceptible ----- In line 191, Several genes in this study (hla, hlb, tst, and pvl) have also been detected in aggressive isolates of S. aureus.

Authors respond: Thank you for your comment. From this sentence “Several genes in this study (hla, hlb, tst, and pvl) also have been detected in aggressive isolates of S. aureus.”, it implies that the prevalence of hlb, tst, and pvl genes in the bacteria isolated from long-tailed macaques in our study had a potential to exhibit enhanced virulence or pathogenicity or have the ability to cause severe infections as well as the aggressive S. aureus from previous studies.

Comment 9: In many places, the bacterial name is not italicized, and it needs to be checked and corrected in the whole manuscript.

Authors respond: Thank you for catching this. We have accepted and followed the reviewer’s comment. We checked and corrected the whole manuscript except the term staphylococcal which is not italicized.

In addition, we labeled the word “Maha Sarakham” in the map of Figure 1 as well for more clarification.

On behalf of all the authors, we would like to express our sincere gratitude to the reviewers for their valuable time, insightful suggestions, and constructive comments, which have significantly contributed to enhancing the quality of our manuscript. Your input has been instrumental in improving our work, and we deeply appreciate your dedicated effort with this reviewg.

My best regards.

Yours sincerely,

Natapol Pumipuntu, DVM, Ph.D.

15/7/2023

Reviewer 3 Report

Pumipuntu et al present a report on their molecular characterization of Staphylococcus aureus (both methicillin-resistant and methicillin-susceptible strains) and Staphylococcus argenteus among macaques at a reserve in Thailand. The study is very important, given the clinical significance of Staphylococci in both human and animal health, their high zoonotic potential, and their high capacity for antimicrobial resistance, especially, methicillin resistance. The sparsity of AMR surveillance among animals and the limited molecular epidemiological data on Staphylococci among wildlife, particularly in Thailand, give the study further importance and speaks to its novelty. The background information provided by the authors is sufficient (although too long) and the methods used have been clearly articulated. Furthermore, their conclusions stem from their results, which also satisfy the study objectives. The references cited are also appropriate, and the figures and tables are clear and easy to interpret. Additionally, the manuscript is well written, and the presentation is very clear. Specific comments on the few identified areas for improvement are as follows:

1. The title needs to be shortened to make its understanding clearer.

2. The Introduction section is too long, and needs to be made more concise.

3. The authors need to check the unit in which the temperatures are presented, for example, on Line 137.

Minor English edits are required.

Author Response

Tropical Medicine and Infectious Disease

-------------------------------------------------------------------------------------------------------------------------

Title: Molecular characterization of methicillin-resistant Staphylococcus aureus, methicillin-susceptible Staphylococcus aureus, and Staphylococcus argenteus isolated from free-ranging long-tailed macaques at Kosumpee Forest Park, Maha Sarakham, Thailand

Manuscript Number: tropicalmed-2490476

Author: Natapol Pumipuntu, Tawatchai Tanee, Pensri Kyes, Penkhae Thamsenanupap, Apichat Karaket, Randall C. Kyes

-------------------------------------------------------------------------------------------------------------------------

Response to reviewers

Reviewer # 3 comments:

Pumipuntu et al present a report on their molecular characterization of Staphylococcus aureus (both methicillin-resistant and methicillin-susceptible strains) and Staphylococcus argenteus among macaques at a reserve in Thailand. The study is very important, given the clinical significance of Staphylococci in both human and animal health, their high zoonotic potential, and their high capacity for antimicrobial resistance, especially, methicillin resistance. The sparsity of AMR surveillance among animals and the limited molecular epidemiological data on Staphylococci among wildlife, particularly in Thailand, give the study further importance and speaks to its novelty. The background information provided by the authors is sufficient (although too long) and the methods used have been clearly articulated. Furthermore, their conclusions stem from their results, which also satisfy the study objectives. The references cited are also appropriate, and the figures and tables are clear and easy to interpret. Additionally, the manuscript is well written, and the presentation is very clear. Specific comments on the few identified areas for improvement are as follows:

Comment 1: The title needs to be shortened to make its understanding clearer.

Authors respond: We have accepted the reviewer’s comment. We adjusted the title of this manuscript from “Molecular characterization of methicillin-resistant Staphylococcus aureus, methicillin-susceptible Staphylococcus aureus, and Staphylococcus argenteus isolated from free-ranging long-tailed macaques at Kosumpee Forest Park, Maha Sarakham, Thailand” to “Molecular characterization of Staphylococcus aureus complex isolated from free-ranging long-tailed macaques at Kosumpee Forest Park, Maha Sarakham, Thailand”

Comment 2: The Introduction section is too long, and needs to be made more concise.

Authors respond: We have accepted the reviewer’s comment. We have shorten the Introduction and tried to make it more concise. .

Comment 3: The authors need to check the unit in which the temperatures are presented, for example, on Line 137.

Authors respond: We have accepted the reviewer’s comment. We checked and corrected the unit of temperatures throughout the revised manuscript

In addition, we labeled the word “Maha Sarakham” in the map of Figure 1 as well for more clarification.

On behalf of all the authors, we would like to express our sincere gratitude to the reviewers for their valuable time, insightful suggestions, and constructive comments, which have significantly contributed to enhancing the quality of our manuscript. Your input has been instrumental in improving our work, and we deeply appreciate your dedicated effort with this reviewing.

My best regards.

Yours sincerely,

Natapol Pumipuntu, DVM, Ph.D.

15/7/2023
